# The Plasma DIA-Based Quantitative Proteomics Reveals the Pathogenic Pathways and New Biomarkers in Cervical Cancer and High Grade Squamous Intraepithelial Lesion

**DOI:** 10.3390/jcm11237155

**Published:** 2022-12-01

**Authors:** Sai Han, Junhua Zhang, Yu Sun, Lu Liu, Lingyu Guo, Chunru Zhao, Jiaxin Zhang, Qiuhong Qian, Baoxia Cui, Youzhong Zhang

**Affiliations:** Department of Obstetrics and Gynecology, Qilu Hospital of Shandong University, Jinan 250012, China

**Keywords:** DIA, cervical cancer, HSIL, ELISA, biomarkers, pathogenic pathways

## Abstract

Objective: The process of normal cervix changing into high grade squamous intraepithelial lesion (HSIL) and invasive cervical cancer is long and the mechanisms are still not completely clear. This study aimed to reveal the protein profiles related to HSIL and cervical cancer and find the diagnostic and prognostic molecular changes. Methods: Data-independent acquisition (DIA) analysis was performed to identify 20 healthy female volunteers, 20 HSIL and 20 cervical patients in a cohort to screen differentially expressed proteins (DEPs) for the HSIL and cervical cancer. Subsequently, Gene Ontology (GO) and Kyoto Encyclopedia of Genes and Genomes (KEGG) analyses were used for functional annotation of DEPs; the protein–protein interaction (PPI) and weighted gene co-expression network analysis (WGCNA) were performed for detection of key molecular modules and hub proteins. They were validated using the Enzyme-Linked Immunosorbent Assay (ELISA). Results: A total of 243 DEPs were identified in the study groups. GO and KEGG analysis showed that DEPs were mainly enriched in the complement and coagulation pathway, cholesterol metabolism pathway, the IL-17 signaling pathway as well as the viral protein interaction with cytokine and cytokine receptor pathway. Subsequently, the WGCNA analysis showed that the green module was highly correlated with the cervical cancer stage. Additionally, six interesting core DEPs were verified by ELISA, APOF and ORM1, showing nearly the same expression pattern with DIA. The area under the curve (AUC) of 0.978 was obtained by using ORM1 combined with APOF to predict CK and HSIL+CC, and in the diagnosis of HSIL and CC, the AUC can reach to 0.982. The high expression of ORM1 is related to lymph node metastasis and the clinical stage of cervical cancer patients as well as the poor prognosis. Conclusion: DIA-ELSIA combined analysis screened and validated two previously unexplored but potentially useful biomarkers for early diagnosis of HSIL and cervical cancer, as well as possible new pathogenic pathways and therapeutic targets.

## 1. Introduction

Cervical cancer is the most common gynecological malignant tumor in women. According to the latest cancer research data released by WHO, the number of new cases of cervical cancer in the world in 2020 was about 600,000, and the number of death cases was about 340,000 [1]. Although the etiology of cervical cancer is relatively clear, it is caused by the persistent infection of high-risk human papillomavirus (HPV). However, 90% of HPV infected patients can automatically clear the HPV virus in the body within 2 years [2], and the progress from HPV persistent infection to cervical cancer is slow, suggesting that HPV infection is a necessary condition for the occurrence and development of cervical cancer, but not a sufficient condition. A large number of studies have shown that many factors such as imbalance of intestinal flora, regulation of non-coding RNA and exosomes, abnormal methylation and single nucleotide polymorphisms, and disruption of cervical immune microenvironment are involved in the occurrence and development of cervical cancer [3,4,5,6,7]. Similarly, these factors also play key roles in carcinogenesis among other gynecologic malignant tumors, such as the complex regulatory network formed by the non-coding RNA in the endometrial cancer [8,9,10,11].

At present, the diagnosis of cervical cancer and cervical intraepithelial neoplasia relies on the three-steps method of HPV/cytology (TCT), colposcopy and biopsy, which is time-consuming and costly. In most developing countries, there are shortages of cytological, colposcopy and histological diagnostic physicians, which lead to the risk of missed diagnosis and overdiagnosis of cervical cancer and HSIL. Serological detection is an objective and simple diagnostic method. At present, opportunities for the serological diagnosis of cervical cancer and HSIL are scarce. Squamous cell carcinoma antigen (SCC-Ag) is a serological detection method for locally advanced cervical cancer, but it has poor sensitivity and specificity, and has no clinical significance in the diagnosis of HSIL.

Nowadays, data-independent acquisition (DIA)-based quantitative proteomics technology is a useful technology for large-scale protein identification and quantification. Compared to the traditional mass spectrometry (MS) method in data dependent acquisition (DDA) mode, DIA has the advantages of deep proteome coverage, high quantitative reproducibility and accuracy. It has been applied in biomarker discovery, clinical research, fundamental investigation, and other fields [12,13,14]. In this work, we performed the DIA-method mentioned above to provide a comprehensive protein profile from the plasma to further reveal the pathogenesis process of HSIL and cervical cancer. Besides, our study also aims to screen candidate biomarkers that can identify HSIL and cervical cancer patients.

## 2. Materials and Methods

### 2.1. Study Design and Collection of Clinical Samples

The study design is shown in Figure 1. A total of 125 clinical plasma samples, from 30 healthy female volunteers (named CK), 41 HSIL (named HSIL) and 54 cervical cancer patients (named CC), which were divided into the discovery group and validation group, were collected from the Qilu Hospital of Shandong University from January 2019 to June 2020. Among the specimens, 20 CK, 20 HSIL and 20 CC specimens were analyzed by the DIA method to investigate the protein alterations in these three states; these 60 specimens were a subset of the discovery group. Additionally, ELSIA analysis of 30 CK, 41 HSIL and 54 CC were performed to validate the selected proteins. These 125 specimens were a subset of the discovery group, and were included in the validation group.

For plasma collection, blood samples from all patients were obtained before breakfast on the second day after hospitalization. The samples were immediately processed according to the standardized protocol recommended by the HUPO Plasma Proteome Project [15]. Briefly, blood was collected into plastic K2EDTA tubes (BD), manually inverted 10 times and centrifuged at 2000 r/min for 10 min at 4 °C, and then stored in aliquots at −80 °C. Meanwhile, 10 µl of serum from each sample was mixed as a QC sample. All the clinical information of the patients was collected from the hospital’s His system.

### 2.2. Sample Preparation and Fractionation for DDA Library Generation

Based on the relevant operating instructions, the proteins rich in expression in plasma were separated and processed through the human 14 multiple affinity removal system column to determine the high and low abundance proteins, and then desalted and concentrated with a 5 kDa ultrafiltration tube (Sartorius, Goettingen, Germany). Next: add SDT buffer solution (4% SDS, pH 8.0), heat to 100 °C for 15 min, and then centrifugate at high speed for 20 min. After the supernatant is taken out, quantitative analysis is carried out with BCA kit (Bio Rad, Hercules, CA, USA), and the remaining samples are stored at −80 °C.

### 2.3. Data Dependent Acquisition (DDA) Mass Spectrometry Assay

During this experiment, all fractions used to generate the DDA library were detected by Thermoscience QExactive HF X mass spectrometer. Next: take the peptide (1.5 μg). Load it onto EASY Trapcolumn (Thermo Scientific, P/N 164946, Waltham, MA, USA), and then separate the sample on the corresponding analytical column (ES802, 2 um, 75 um × 25 cm). Control the concentration gradient of buffer B to 250 nl/min, 120 min. During MS detection, the parameters set are as follows: scanning range 300–1800 m/z, resolution 60,000, AGC 3e6, and maximum IT 25 ms. After scanning ddMS2 for 20 times, start a complete MS–SIM scan. The parameters set during MS2 scanning are as follows: the resolution is 15,000, AGC is 5e4, and the maximum IT is 25 ms.

### 2.4. Mass Spectrometry Assay for Data Independent Acquisition (DIA)

In the process of sample analysis, LC-MS/MS analysis is carried out for peptides in each sample based on the data independent acquisition (DIA) mode. MS-SIM scanning is required for each cycle. The scanning parameters are set as follows: the resolution is 120,000 under full scanning conditions: AGC 3e6; Maximum IT 50 ms. The parameters set in DIA scanning mode are as follows: AGC target 3e6; Max IT auto; Continuous operation for two hours. According to relevant experimental requirements, set the linear gradient of buffer B to 250 nl/min. During the experiment, after every 6 injections at the beginning of MS research, select the DIA mode (divide the sample equally) when injecting the sample, and judge the MS performance based on the results obtained.

### 2.5. Mass Spectrometry Data Analysis

For DDA database data, Spectronaut TM14 software was selected to search for FASTA sequence in order to meet relevant application requirements during processing, and the data meeting the requirements were sorted out. Uniprot human database was selected to add iRT peptide sequence. The relevant DIA data is searched through Spectraut TM14, and the relevant spectral library is established: set the parameters of the software according to the application requirements: dynamic iRT, enable MS2 level correction interference, and filter all results according to the Q cut-off value of 0.01

### 2.6. Bioinformatical and Statistical Analysis

#### 2.6.1. Subcellular Localization

CELLO (http://cello.life.nctu.edu.tw/ (accessed on 29 June 2013) is a commonly used classification system. In this paper, we mainly analyze the subcellular localization of proteins through CELLO.

#### 2.6.2. Domain Annotation

Search through InterProScan software (EMBL-EBI, Cambridgeshire, UK) to obtain relevant sequence information and judge the characteristics of proteins in Pfam based on the results.

#### 2.6.3. GO Annotation

In the analysis of differentially expressed proteins, NCBI BLAST+client software is used to determine the sequence information of these proteins, and then search to obtain relevant homologous sequences.

#### 2.6.4. KEGG Annotation

Follow the annotation steps to search in the KEGG library, identify the properties of proteins, and map to the path in KEGG based on the above processing, so as to meet the application requirements.

#### 2.6.5. Enrichment Analysis

In the enrichment analysis process, based on Fisher’s exact test, all quantitative proteins are set as the background data set, and the derived *p* value is corrected by the Benjamin system that has been tested many times to meet the subsequent application requirements. When the *p* value is found to be lower than 0.05, it can be considered that the relevant functional categories meet the requirements.

#### 2.6.6. Protein-Protein Interaction Analysis

According to the relevant research requirements, search the IntAct database to determine the protein-protein interaction (PPI) information, and then use the relevant gene symbols to describe the results. The final results are downloaded in the XGMML format and sent to the Cytoscape software (Cytoscape Team, San Diego, CA, USA) to visualize the results, and establish an intuitive protein interaction network to support subsequent analysis. At the same time, we also need to calculate the weight of each protein, so as to clarify its position in the PPI network.

#### 2.6.7. WGCNA Analysis (Weighted Gene Co-Expression Network Analysis)

The WGCNA in R package (Version 1.69) (R Core Team, Vienna, Austria) is mainly used to identify different protein modules. In the process of generating related protein co expression networks, the log2 abundance sample matrix is selected according to relevant standards and requirements.

#### 2.6.8. ELISA

The plasma samples for the detection of the selected proteins were kept frozen at −80 °C until use. The ELISA kits were obtained from Meimian Biotechnology (Yancheng, Jiangsu, China) and used according to the manufacturer’s recommendations.

#### 2.6.9. Statistical Analysis

GraphPad Prism version 5.01 (GraphPad Software Inc., San Diego, CA, USA) was used for statistical analysis. In the present study, data are expressed as the means with standard deviations (SDs), and statistical comparisons were performed using Student’s *t*-test or ANOVA analysis. Chi-square test and Kaplan–Meier analysis were performed to explore the relationship between molecular expression and the clinicopathological factors as well as the survival prognosis. *p* < 0.05 was considered to indicate a statistically significant result.

## 3. Results

### 3.1. An Overview of the Quantitative Proteomics Analysis

In our study, 12915 peptides corresponding to 1357 proteins were identified in the spectral library, and a total of 1096 proteins and 6746 peptides were detected in our samples (Appendix A Appendix A). To assess the quality of the proteomic data, quality control (QC) analysis was performed. Our Appendix A show that there are enough data points for each peak and complete peak area integration to ensure the accuracy of DIA quantitative results; the peak capacity can ensure that each peak has a good resolution (Appendix A Appendix A). Enough iRT points were detected and the retention time of each sample has a certain stability (Appendix A Appendix A). Our quality control samples (QC) showed a strong correlation (DIA sequencing data is eligible > 0.9) (Appendix A Appendix A). All the results show that the DIA sequencing data is eligible.

The Venn diagram in Figure 2A shows the number of identified proteins displaying significant quantitative similarities and differences among the three groups. There were 1080, 1070, and 1066 proteins identified in CK, HSIL, and CC, respectively. A total of 1021 proteins were shared by all three groups, demonstrating that a large set of overlapping proteins (75.24%) was detected, which validated the robustness of the proteome maps to some extent. Additionally, principal component analysis (PCA) using detected proteins grouped by the three states (Figure 2B) revealed not ideal differences, which may be related to the plasma sample itself. The DIA identification and quantification results and heatmap are also shown in Figure 2C,D.

### 3.2. Identification of Differentially Expressed Proteins

We set up three comparison groups: CC vs. CK; CC vs. HSIL; HSIL vs. CK. For the three groups’ comparisons by one-way ANOVA, proteins with a fold change (FC) ≥ 1.5 or ≤ 0.67 and *p*-value < 0.05 were defined as significantly differentially expressed proteins (DEPs). In total, 243 DEPs of three comparison groups were identified in the present study (Figure 3A, Appendix A Appendix A). When analyzing the DEPs, we found that the CC group had a total of 81 DEPs including 51 upregulated proteins and 30 downregulated proteins, compared with CK (Figure 3A,B; Appendix A Appendix A). There were 60 DEPs in the CC group compared with the HSIL group, including 13 upregulated proteins and 47 downregulated proteins (Figure 3A,C; Appendix A Appendix A). There were 102 DEPs in the HSIL group compared with the CC group, including 91 upregulated proteins and 11 downregulated proteins (Figure 3A,D; Appendix A Appendix A). The data above suggested the potential for finding suitable diagnostic biomarkers for diagnosis of HSIL and cervical cancer in our study cohort.

### 3.3. Functional Annotation Analysis of the DEPs Related to the HSIL and Cervical Cancer Occurrence

To further understand the function of the proteins related to disease progress in HSIL and cervical cancer, the subcellular localization and domain analysis, the gene ontology (GO), Kyoto encyclopedia of genes and genomes (KEGG) pathway enrichment as well as the protein-protein Interaction Networks (PPI) analysis were performed. Pairwise comparisons among the three groups are provided in Appendix A. Here we only show the results of the ANOVA analysis between the three groups. The subcellular localization analysis showed that the extracellular protein was most abundant (Figure 4A). The domain analysis showed that the largest change in the amount was trypsin, and the largest fold change was the coagulation Factor Xa inhibitory site (Figure 4B). The main significantly enriched GO terms of the biological process (BP) included response to stimulus, biological regulation, and metabolic process. Binding, catalytic activity, and molecular function regulator were the main enriched GO terms of molecular function (MF). Moreover, the main representative GO terms of the cellular component (CC) were extracellular region (Figure 5A; Appendix A Appendix A). KEGG pathway enrichment results showed that the complement and coagulation cascades, staphylococcus aureus infection, cholesterol metabolism and so on, were the most significantly enriched (Figure 5B; Appendix A Appendix A).

### 3.4. Protein-Protein Interaction (PPI) Network and Weighted Gene Co-Expression Network Analysis (WGCNA)

To study the molecular mechanism of DEPs from a systematic perspective, we constructed a PPI network to explore the relationship between the proteins. The 243 DEPs were submitted to the STRING 11.0 database via Cytoscape 3.8.0 software to obtain the PPI network diagram (Figure 6). We screened out a series of core proteins such as ORM1, with 18 edges. Moreover, QSOX1, SERPINC1, APOC3 etc. were the other hub proteins with no less than 10 edges. 

WGCNA was performed to reveal protein modules related to the HSIL and cervical cancer stages [16,17]. As a result, six modules with different expression patterns were identified (Figure 7A,B). Based on the criteria (Pearson r ≥ 0.3 or r ≤ −0.3, *p*-value ≤ 0.05), we found that in the cervical cancer stage, the green module of 73 proteins was highly correlated with it. The co-expressed proteins network of the green module was visualized, of which several hub proteins were identified (Figure 7C,E). For example, the core proteins such as FN1 and APOF were involved in the process of cervical cancer. The grey module of 431 proteins was also weakly correlated with the cervical cancer stage, and a series of core proteins such as ORM1 and QOSX1, represented the hub protein (Figure 7D,F). Then KEGG function enrichment analysis of the green module showed that complement and coagulation cascades were the main enriched pathways in the cervical cancer stage (Appendix A Appendix A). The GO terms showed that the response to stimulus is the most important in BP. Binding is the key point in MF, and the extracellular region is the most abundant part in the CC (Appendix A Appendix A). As for the HSIL, the results of WGCNA showed that there was no apparent correlation between the six expression patterns with this stage.

We also performed the PPI and WGCNA analysis in the cervical cancer group. According to the nine clinical clinicopathological factors and the six modules as well as the criteria above, we found that the HPV infection was correlated with the blue and grey module, the differentiation stage was related to the brown and turquoise module, the tumor size was correlated with the grey module, and the lymph vascular space involvement (LVSI) was correlated with the green module (Appendix A Appendix A). Therefore, we summarized the specific information such as KEGG and GO information in Appendix A.

### 3.5. Validation of the Proteomic Data by ELSIA

To verify the expression patterns of the proteins identified by DIA technology, six interesting core DEPs were verified by ELISA, which reflected their expression at the protein level in the blood plasma. ELISA assay was performed in 30 healthy volunteers, 41 HSIL and 54 cervical cancer patients. Our results indicated that APOF and ORM1 showed nearly the same expression pattern between the relative expression pattern in DIA and ELISA, and they all belonged to the green and grey module of WGCNA (Figure 8A,B; Appendix A Appendix A). The results (Figure 8A) revealed that the plasma expression level of ORM1 has the highest expression in HSIL, followed by cervical cancer, and the lowest expression in healthy volunteers (1092 ± 7.3 vs. 952.2 ± 6.9 vs. 859.2 ± 7.4 ng/mL, *p* < 0.001). The APOF in CC patients was higher than that of the HSIL (415.4 ± 4.6 vs. 402.2 ± 4.5 ng/mL; *p* = 0.0485) and CK (415.4 ± 4.6 vs. 364.2 ± 10.8 ng/mL; *p* < 0.001). APOF in HSIL patients was also higher than CK (*p* = 0.0006) (Figure 8B). The receiver operating characteristic curve (ROC) is presented in Figure 8C,D, and the area under the curve (AUC) was 0.978 by using ORM1 combined with APOF to predict CK and HSIL + CC; in the diagnosis of HSIL and CC, the AUC can reach to 0.982. The AUC in these two groups was similar by detecting ORM1 alone, but the AUC of APOF alone in these two groups was not ideal. These data demonstrated that ORM1 with or without APOF has the potential to become a diagnostic indicator of HSIL and cervical cancer.

### 3.6. Relationship between ORM1/APOF Expression and the Clinicopathological Factors as well as Survival Prognosis

We performed Chi-square test and Kaplan–Meier analysis in the cervical cancer patients, to explore the relationship between ORM1/APOF expression and the clinicopathological factors as well as the survival prognosis. Some patients did not undergo subsequent surgery or were lost to follow-up, so a total of 43 cervical cancer patients underwent Chi-square test and 35 patients underwent Kaplan–Meier analysis. The cervical cancer patients with the top 50% expression were defined as high expression and the others as low expression. The results are listed in Table 1 and Figure 8E,F. We found that the expression level of ORM1 in cervical cancer tissues was not correlated with age, histology, tumor differentiation, clinical stage, tumor size, LVSI or deep stromal invasion (DSI), but was related to lymph node metastasis (LN) and clinical stage. However, neither of the clinicopathological factors above were correlated with the expression of APOF. As for the survival analysis, neither ORM1 nor APOF expression were correlated with the PFS progression free survival (PFS) and overall survival (OS) (Appendix A Appendix A). However, when we defined the top 20% expression as high expression and the remaining 80% as low expression, the ORM1 was related with the PFS and OS of these cervical cancer patients (Figure 8E,G).

## 4. Discussion

The mechanisms of normal cervix changing into HSIL and invasive cervical cancer are still not completely clear. Multiple factors including the HR-HPV persistent infection are involved in this long carcinogenic process. To identify genes carrying genetic information, scientists have used whole-genome sequencing and high-throughput viral integration detection technology to provide insights into HPV integration-driven cervical carcinogenesis [18]. However, proteins are still the final executors of the genes and mRNAs. They may reflect the influence of genome and environmental factors accurately and dynamically, so many studies have explored the differentially expressed proteins in the normal cervix, cervical intraepithelial neoplasia (CIN) and cervical squamous cell carcinoma (CSCC) tissues by differential proteomics technique. They found that S100A9, eEF1A1, PKM2, COPA and so on, may become candidate markers for early diagnosis of cervical cancer and new targets for therapy [19,20]. To our knowledge, this is the first study using the DIA of plasma to investigate protein expression profiles of cervical disease. Because of the inconsistent proportions of lesion components in the tissue especially in the HSIL, we believe that in the plasma, the results would be more objective.

In our study, we present a cervix-specific spectral library that comprises 12,915 peptides and 1357 proteins from proteotypic peptides, which largely expands our understanding of proteins expressed in HSIL and cervical cancer. We mainly used two approaches: expression difference analysis and functional analysis. Because the plasma protein is a class of compounds with the largest content, complex composition and wide range of functions in plasma solid components, the protein expression pattern as well as their function were different from the results of previous cervical tissue protein sequencing data [19,20]. In total, we found 243 DEPs, and the hub proteins identified from the PPI such as ORM1, APOF, QSOX1 and so on; most of them have not been reported before in cervix disease, which undoubtedly provides us with new research targets. In the functional analysis, we are particularly concerned about the changes in the relevant molecular pathways.

The KEGG analysis showed that the IL-17 signaling pathway has a significant change in CC vs. CK and HSIL vs. CK groups (Appendix A Appendix A), which is consistent with most reports that the IL-17 pathway plays important roles in cervical cancer and HSIL [21,22,23]. The viral protein interaction with cytokine and cytokine receptor pathway also has an outstanding performance in CC vs. CK and CC vs. HSIL groups (Appendix A), which is consistent with the fact that the etiology of cervical cancer is associated with persistent high-risk HPV infection. Moreover, the ANOVA analysis showed that the complement and coagulation pathway of the immune systems changed in all three groups. This suggests that immune factors may play an important role in the occurrence and development of cervical cancer. More and more studies have focused on the antitumor immunity by targeting the inhibition of coagulation factor, and we can learn from this idea in cervical disease [24,25,26]. In addition, the cholesterol metabolism pathway was also on the list in CC vs. HSIL and HSIL vs. CK groups. This is also in line with our previous studies of related pathway molecules such as LRP11 and other studies [27,28].

Interestingly, in the HSIL vs. CK group (Appendix A Appendix A), neurodegenerative disease pathways such as Alzheimer’s disease, Huntington disease, Parkinson’s disease and so on have significant changes; there was only one article hypothesizing whether HPV may be linked to Alzheimer’s disease [29]. Our DIA protein sequencing results provide more direct evidence that warrants further investigation.

SCC-Ag is used clinically to predict squamous cell carcinoma (SCC). Many studies had shown that the serum SCC-Ag was consistently associated with recurrence and survival of newly diagnosed cervical cancer [30,31], and it can predict the survival outcomes of patients with cervical cancer after radiotherapy [32]. However, it lacks specificity for cervical cancer and also lacks predictability in lymph node metastases [33]. Some studies had discovered and validated novel biomarkers for detection of cervical cancer. However, most of them were derived from histological bioinformatics analysis [34]. Therefore, identification of specific blood biomarkers remains an urgent need for increasing the diagnostic accuracy. In general, the prevailing diagnostic model after DIA is the verification of parallel reaction monitoring (PRM) [35,36]. However, PRM is expensive and it is not suitable for clinical testing. In our study, instead of PRM, we used ELISA technology and validated ORM1, APOF, QSOX1, TTN, GABRG2, F10 using ELISA (Appendix A Appendix A); we found that the results of APOF and ORM1 have high consistency with DIA sequencing. ORM1 with or without APOF has high sensitivity and specificity in diagnosing HSIL and cervical cancer. The high expression of ORM1 is related to lymph node metastasis and clinical stage of cervical cancer patients as well as the prognosis, which is undoubtedly an important supplement to SCC-Ag.

Actually, ORM1, or alpha-1 acid glycoprotein (AGP), which acts an acute response protein with multiple modulating activities, constitutes 1% to 3% of plasma proteins in humans and is mainly synthesized in the liver, has become a famous biomarker in many tumors such as lymphoma, bladder cancer, breast cancer, gastric cancer, as well as epithelial ovarian cancer [37,38,39,40,41]. The role of ORM1 as a biomarker of cervical disease is reported for the first time in our study. Its function includes modulating immunity, binding and carrying drugs, maintaining the barrier function of capillary, and mediating the sphingolipid metabolism [42], but no study was found for ORM1 in cervical cancer. In the future, the ORM1 function research in cervical disease is warranted. APOF is known as a lipid transfer inhibitor protein, given its ability to inhibit cholesteryl ester transfer protein-mediated transfers of cholesteryl esters and triglycerides. Its role as a biomarker was mainly reflected in digestive system tumors such as hepatocellular carcinoma and cholangiocarcinoma [43,44], and only one article reported that the APOF levels were more than twice as high in the E6/E7 mRNA HPV-positive group than HPV-negative in oropharyngeal squamous cell carcinoma (OPSCC) [45], meaning that it can discriminate between HPV-positive and HPV-negative OPSCC. However, there is no research about its functional studies in cervical cancer. In the next step, we may carry out this work.

Persistent infection of high-risk HPV is the main cause of cervical cancer and HSIL, so plasma proteomic sequencing and verification of these patients is also very important. However, due to the lack of access to the plasma samples of these patients, there was no data for the persistent infection of high-risk HPV in our study. In the follow-up work, we will focus on these patients.

## 5. Conclusions

In the present study, we reported the large-scale protein profiles related to HSIL and cervical cancer through the plasma DIA-based quantitative proteomics. Through the functional and WGCNA analysis of the identified 243 DEPs, we found the star pathways involved in the whole pathogenic process such as the complement and coagulation pathway as well as the cholesterol metabolism pathway. Other interesting signaling pathways that have not previously been tested also provided a new insight to study the occurrence and development of the cervical diseases. By validating the hub proteins using the ELISA, we found that ORM1 and APOF could be the new potential plasma biomarkers in HSIL and cervical cancer, and further functional studies are worth exploring.

## Figures and Tables

**Figure 1 jcm-11-07155-f001:**
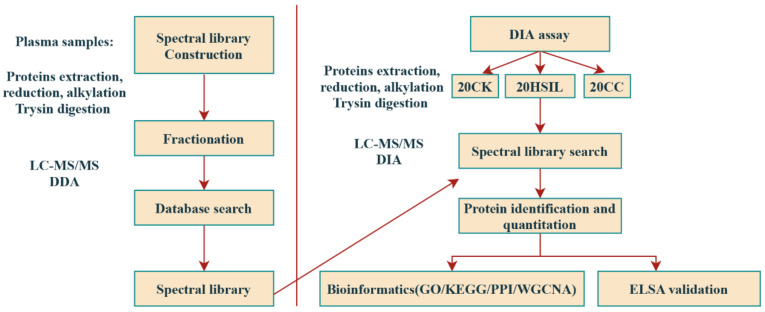
DIA experimental and analytic flow.

**Figure 2 jcm-11-07155-f002:**
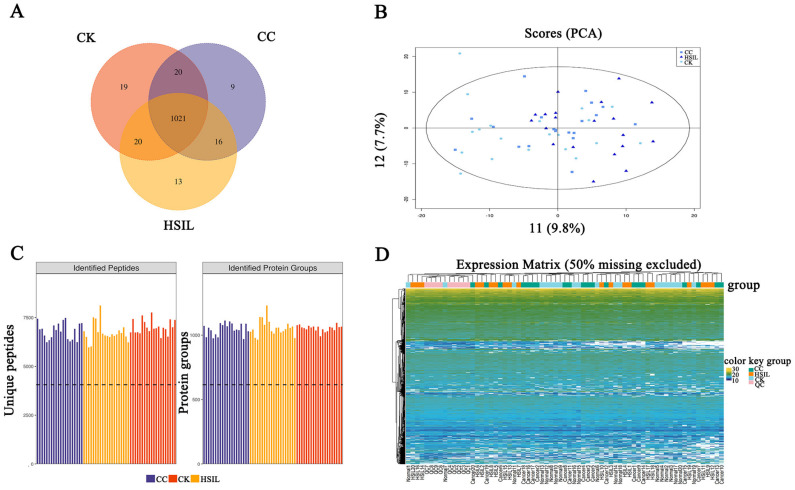
General overview of protein identification. (**A**) Venn diagram showing the number of proteins common and unique to healthy, HSIL and cervical cancer patients; (**B**) PCA score plot. Samples from the CK (n = 20), HSIL (n = 20) and CC (n = 20) groups are plotted along the three principal component. (**C**) The identified peptides and protein of the three groups. (**D**) DIA quantitative heat map of total protein. Protein expression levels are shown in different colors, with yellow indicating a stronger signal, dark blue indicating a weaker signal, and white indicating no quantitative information.

**Figure 3 jcm-11-07155-f003:**
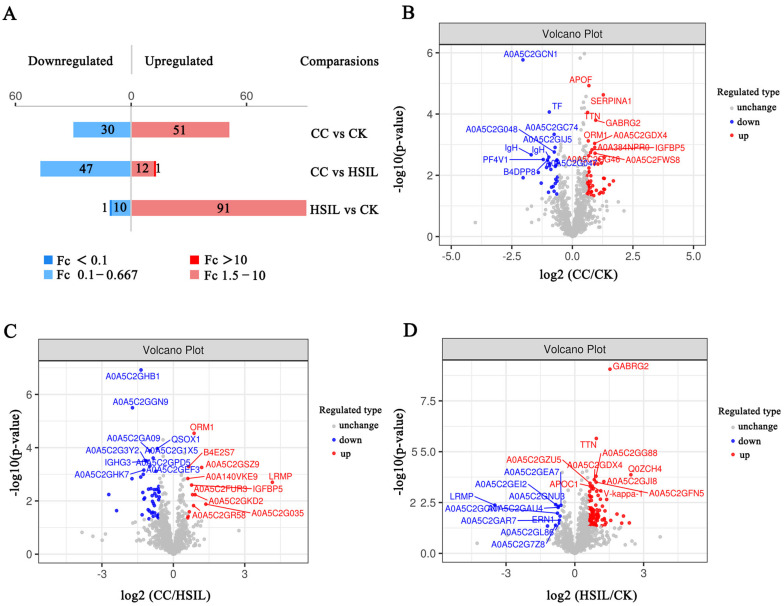
Expression of differential proteins (DEPs). (**A**) Bar chart of protein quantitative difference results; (**B**) Volcano plot of differentially expressed proteins between CC and CK groups; (**C**) Volcano plot of differentially expressed proteins between CC and HSIL groups; (**D**) Volcano plot of differentially expressed proteins between HSIL and CK groups.

**Figure 4 jcm-11-07155-f004:**
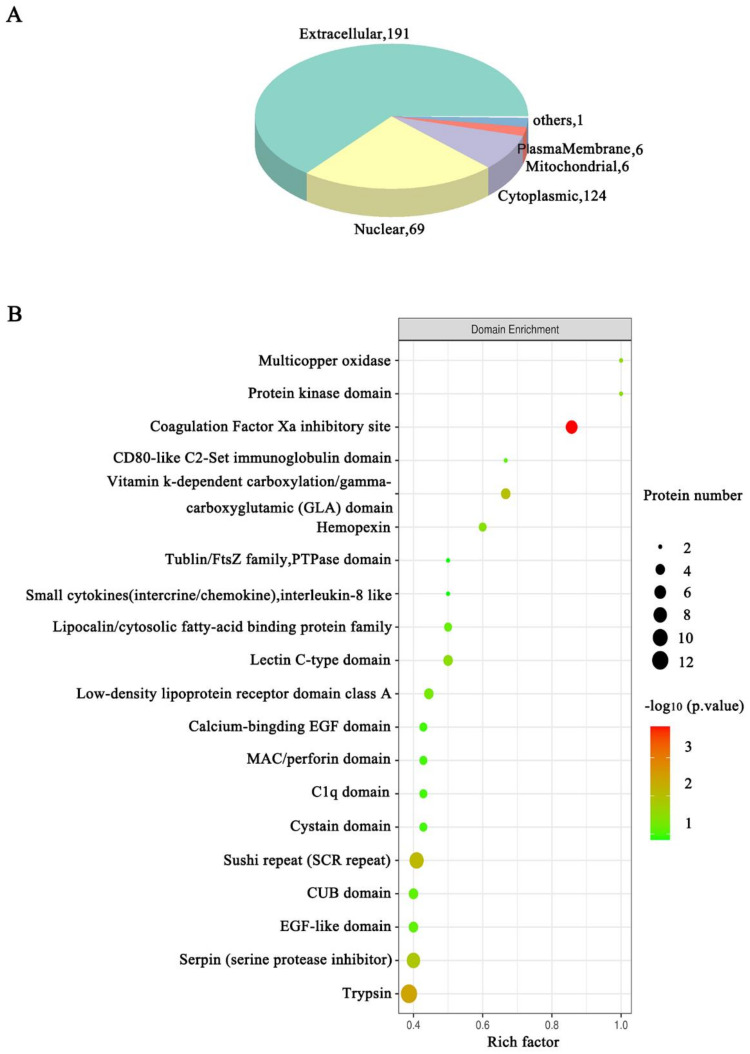
Subcellular localization and domain enrichment analysis of DEPs. (**A**) Pie chart of subcellular localization of DEPs; (**B**) The domain enrichment analysis of DEPs.

**Figure 5 jcm-11-07155-f005:**
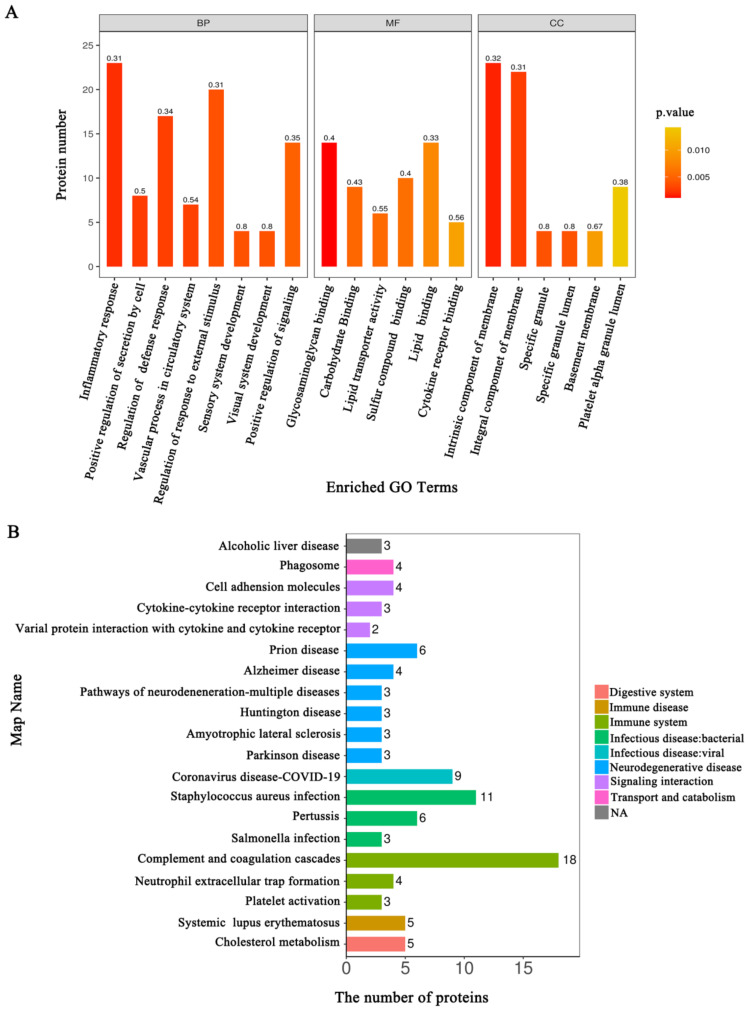
GO and KEGG pathway with top-20 for DEPs between CK, HSIL and CC groups. (**A**) GO analysis: The biological process (BP) categories, molecular function (MF) and cellular component (CC) categories were presented; (**B**) KEGG pathway analysis.

**Figure 6 jcm-11-07155-f006:**
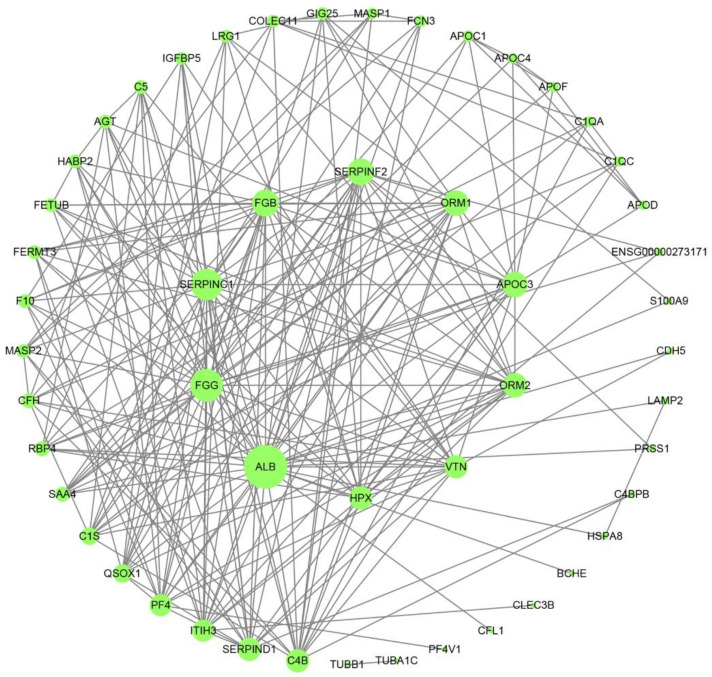
DEPs co-expression network. Nodes are target proteins. The size of the nodes represents the difference in protein expression, and the lines represent the connection degree. The higher the connectivity, the more important it is in the protein-protein interaction (PPI) network.

**Figure 7 jcm-11-07155-f007:**
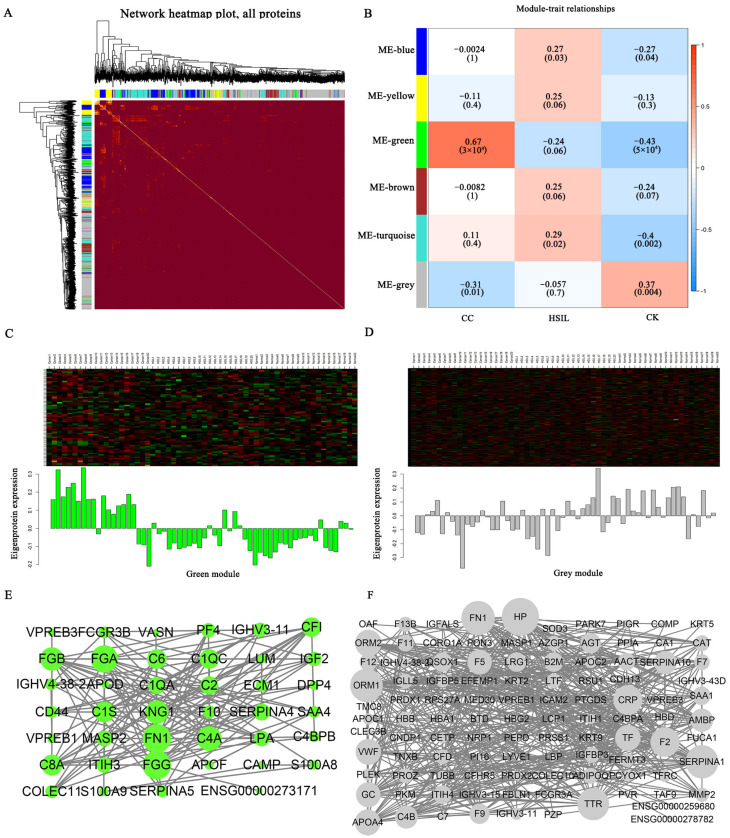
WGCNA analysis of DEPs. (**A**) Heat map of the visualized protein network. The heat map depicts the difference between all DEPs in the analysis. Light colors indicate low overlap and dark colors indicate high overlap. (**B**) A total of six co-expression modules were constructed, and each color represents one module of the protein co-expression network constructed by WGCNA. The top numbers in each cell represent Pearson r, and the bottom numbers represent *p*-values. (**C**) Trend of green module protein expression. (**D**) Trend of gray module protein expression. (**E**) PPI network of green module. (**F**) PPI network of gray module.

**Figure 8 jcm-11-07155-f008:**
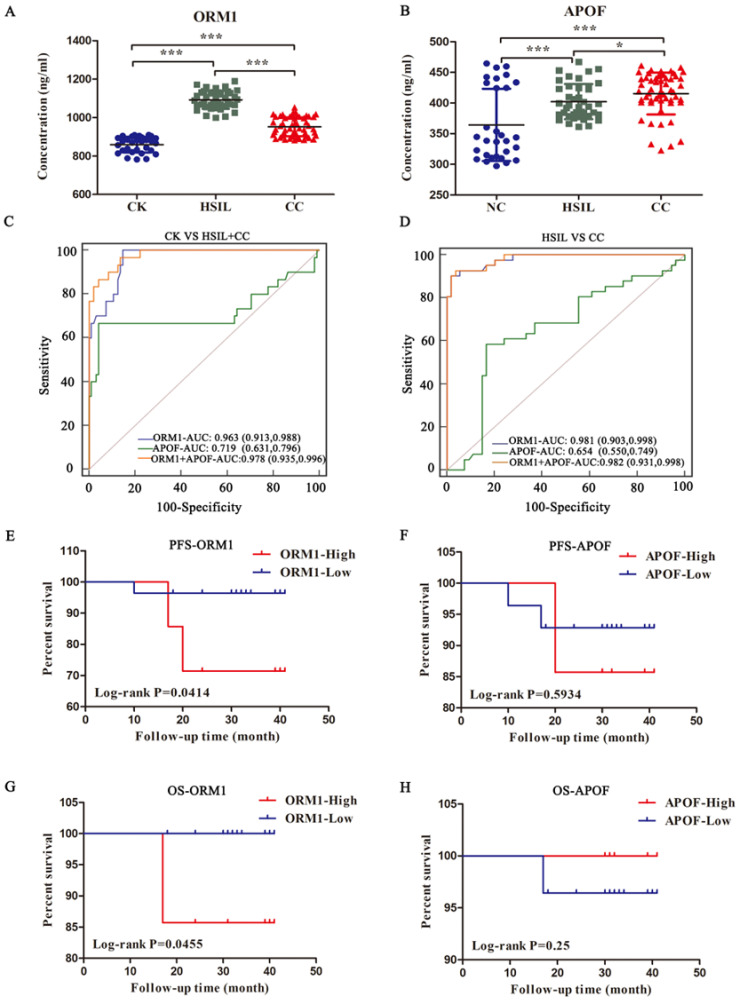
ELSIA verification of proteins identified by DIA analysis. The expression of hub proteins ORM1 (**A**) and APOF (**B**) was measured by ELISA. (**C**) The ROC of ORM1 and APOF in identifying CK and HSIL/CC; (**D**) The ROC of ORM1 and APOF in identifying HSIL and CC. (**E**,**F**) The PFS of ORM1/APOF high expression cervical cancer patients (20%) compared with the ORM1/APOF low expression (80%); (**G**,**H**) The OS of ORM1/APOF high expression cervical cancer patients (20%) compared with the ORM1 low expression (80%).**p* < 0.05; *** *p* < 0.0001.

**Table 1 jcm-11-07155-t001:** Association between ORM1/APOF expression and clinicopathological factors.

Variables	No.	Expression of ORM1			Expression of APOF		
		Low	High	*p*-Value	Low	High	*p*-Value
Age				0.6388			0.8869
≤45	20	9	11		10	10	
>45	23	12	11		11	12	
Histology				0.9667			0.9667
SCC	40	20	20		20	20	
Adenocarcinoma	3	1	2		1	2	
Differentiation				0.6578			0.0946
Low-Moderate	19	10	9		12	7	
Moderate-High	24	11	13		9	15	
Clinical Stage				0.0299 *			0.9374
Stage I-II	34	20	14		16	18	
Stage III-IV	9	1	8		5	4	
Tumor Size				0.7497			0.2117
<4 cm	35	18	17		15	20	
≥4 cm	8	3	5		6	2	
LVSI				0.897			0.5329
Negative	25	12	13		11	13	
Positive	18	9	9		10	8	
LNM				0.0299 *			0.9374
Negative	34	20	14		16	18	
Positive	9	1	8		5	4	
DSI				0.8869			0.8869
<1/2	20	10	10		10	10	
≥1/2	23	11	12		11	12	

***** *p* < 0.05 SCC, squamous cell carcinoma; LNM, lymph node metastasis; LVSI, lymph vascular space involvement; DSI, deep interstitial infiltration. * Statistically significantly value.

## Data Availability

Not applicable.

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
