# Peer review of "The Plasma DIA-Based Quantitative Proteomics Reveals the Pathogenic Pathways and New Biomarkers in Cervical Cancer and High Grade Squamous Intraepithelial Lesion"

_jcm, 2022, doi:10.3390/jcm11237155_

Round 1

Reviewer 1 Report

It is considered to be a meaningful thesis that uses bioinformatics techniques widely used in recent studies to discover factors that are significant for the diagnosis of cervical cancer and verify them with the ELSA method.

However, in this study, although it is known worldwide that HPV infection is the main cause of cervical cancer, the study was not conducted on plasma from positive and negative patients with HPV infection, so the limitations of this study should be explained.

In addition, since it is the first study of cervical cancer using plasma samples, although it is original, further discussion on its usefulness is needed.

Author Response

Response to Reviewer 1 Comments

Point 1: In this study, although it is known worldwide that HPV infection is the main cause of cervical cancer, the study was not conducted on plasma from positive and negative patients with HPV infection, so the limitations of this study should be explained.

Response 1: Thanks for your suggestion. Persistent infection of high-risk HPV is the main cause of cervical cancer and HSIL, so plasma proteomic sequencing and verification of these patients is also very important. However, due to the unaccessibility of the plasma samples of these patients, there is no data for the persistent infection of high-risk HPV in our study. In the follow-up work, we will focus on this work. We have added this explanation in our manuscript.

Point 2: In addition, since it is the first study of cervical cancer using plasma samples, although it is original, further discussion on its usefulness is needed.

Response 2: Thanks for your suggestion. The usefulness of the large-scale protein profiles related to HSIL and cervical cancer through the plasma DIA-based quantitative proteomics has been discussed in the introduction and conclusion part.

Reviewer 2 Report

In this study the authors search for plasma proteins associate with cervical lesions and cervical cancer using MS, and validate a number of their findings using ELISA. The result is two proteins, out of which one (ORM1), replicates.

In general the paper reads well, but the methodology in particular needs some work to be readable.

Specific questions:

1. Did the authors apply any kind of correction for multiple testing in the initial analysis (i.e. identification of DEP)? It not this should be included and will have an effect on the number of DEP candidates.

2. I would like to see a table with the 243 DEPs initially found and the p values (in Supplementary materials).

3. The authors should use standardized proteins nomenclature (i.e. Uniprot number), the first time they bring up a protein. This could also be included in the Supplementary table. For many proteins there are several trivial names and these may be confused.

4. It is not clear what criteria were used to select the DEP for replication? Was it due to fold-difference between groups, of assess to ELISA assays?

5. Which were the 6 DEPs selected for validation/replication? That only 2 of 6 DEPs selected for validation replicated indicates that the selection criteria from the initial analyses were too weak. What here the results of the remaining 4?

6. The main result is the ORM1 association. Given the this has been found to be unregulated in other cancers, what would the potential use of this finding in cervical cancer screening or diagnosis? 

Author Response

Response to Reviewer 2 Comments

Point 1: Did the authors apply any kind of correction for multiple testing in the initial analysis (i.e. identification of DEP)? It not this should be included and will have an effect on the number of DEP candidates.

Response 1: Thanks for your suggestion. In order to ensure the authenticity of the data, we have not conducted any correction for multiple testing in the initial analysis. We think that there is no impact on the quantity of DEP.

Point 2: I would like to see a table with the 243 DEPs initially found and the p values (in Supplementary materials.

Response 2: Thanks for your suggestion. We have created a new table containing 243 DEPs named Supplementary Table 3.

Point 3: The authors should use standardized proteins nomenclature (i.e. Uniprot number), the first time they bring up a protein. This could also be included in the Supplementary table. For many proteins there are several trivial names and these may be confused.

Response 3: Thanks for your suggestion. In order to facilitate the reader's search, we also have added the standardized proteins nomenclature in Supplementary Table 3.

Point 4:  It is not clear what criteria were used to select the DEP for replication? Was it due to fold-difference between groups, of assess to ELISA assays?

Response 4: Thanks for your suggestion. We have mentioned that proteins with a fold change (FC) ≥ 1.5 or ≤ 0.67 and p-value < 0.05 were defined as significantly differentially expressed proteins (DEPs) in the “3.2. Identification of Differentially Expressed Proteins” part. When we pick one of these molecules to make ELSIA,  the criteria is “the bigger the multiple of change and the smaller the P value”. Besides, It is also important to be able to purchase a ELISA kit for this molecule.

Point 5: Which were the 6 DEPs selected for validation/replication? That only 2 of 6 DEPs selected for validation replicated indicates that the selection criteria from the initial analyses were too weak. What here the results of the remaining 4?

Response 5: Thanks for your suggestion. The ELISA results of the remaining 4 DEPs were mentioned in discussion part(line 396).  They are presented as Supplementary Figure 10. Compared to other similar studies, 2 of 6 DEPs selected for validation replicated is also acceptable.

Point 6: The main result is the ORM1 association. Given the this has been found to be unregulated in other cancers, what would the potential use of this finding in cervical cancer screening or diagnosis?

Response 6: Thanks for your suggestion. At present, other widely used markers, such as CA125, are also elevated in multiple diseases. ORM1 is elevated in diagnosed cervical cancer, which is related to lymph node metastasis and the prognosis, which is helpful to judge the patient's condition. In the future, the plasma biomarker testing will combine with HPV and TCT testing or imaging such as B-ultrasound and CT to distinguish them from different cancers, which is also our focus.
